

# STAGE 2.0: Sensitivity Transfer Analysis of Greenhouse Emissions

Peter O. Passenier

Independent HCI Professional

*Correspondence to*: Peter O. Passenier (passenr@ziggo.nl)

**Abstract.** A simulation tool has been realized with which the sensitivity of the Earth climate system to human interference can be studied. The tool is intended to offer users of the software, e.g. students in higher education, support in gaining insight into critical aspects of the climate system and thus counteracting common misconceptions with regard to the functioning of this system. The conceptual design of the tool is based on the paradigm *'learning as experimenting'*, encouraging students to explore climate sensitivity in its various aspects in an active manner. The feasibility of the educational application is demonstrated by exploring possible long-term climate consequences and uncertainties of near-term greenhouse-gas mitigation actions, as an outcome of present-day world-wide negotiations under the UNFCCC umbrella.

## 1 Introduction

The application of simulation models at different scales in general may offer a powerful tool in gaining insight into the functioning of complex systems (so-called *'deeper learning'*) and, directly related, the practicing of this insight by experimenting with these models. One of the areas in which this insight is of prime importance concerns our planet itself: the at present still unlimited exploitation of fossil fuels to meet the growing human energy demand and the consequences this will eventually have for the Earth climate system. At the basis stand different technical, 'natural' and behavioral processes, each with their own characteristics. With respect to the last category, a number of 'biases' can be discerned which can be explained from common misconceptions with regard to the functioning of the climate system and the different spatial and temporal scales involved. Thus, partly based on complex computer-simulation runs until the end of the current century, 'safe levels' of greenhouse-gas concentrations are determined (IPCC, 2013) which however, from a paleoclimatic perspective, on the longer term may lead to a large-scale destabilization of the climate system (NASA, Hansen et al., 2008). Others, who belong to the self-declared so-called 'critical school', take observations of the average global temperature over the past decade as sufficient to announce a stabilization, if not a cooling of global climate, an incentive for 'business as usual' with respect to future emission targets for greenhouse gases. An important factor in all this is the notion of *'climate sensitivity'* and the corresponding distinction which has to be made between the different time scales on which *'actual warming'* and *'committed warming'* come into play. The present study describes the realization of an educational simulation tool (STAGE: 'Sensitivity Transfer Analysis of Greenhouse Emissions'), which offers students the possibility to explore climate sensitivity in its various aspects in an active manner.



## 2 Setting the stage

For the conceptual design of STAGE the paradigm *'learning as experimenting'*, with the generic elements of the basic learning cycle ('predict', 'experiment', 'discuss'), is taken as a starting point. As stated in the introduction, the main objective is to encourage students to explore climate sensitivity to 'get a feel' for both 'short-term' (current century) and

possible 'long-term' (beyond) consequences of greenhouse-gas mitigation measures.

As a first step, in Section 3 the classical notions of climate sensitivity and feedback, as used to characterize present-day climate models, are treated from a broader (paleoclimatic) perspective. This will serve as a basis for the STAGE simulation set-up as described in Section 4. For this purpose the simulation core of a previously developed tool for the transient analysis of greenhouse-gas emissions is extended to incorporate long-term effects (STAGE 2.0).

Subsequently, in Section 5 we demonstrate the educational application to the exploration of possible climate consequences and uncertainties of near-term greenhouse-gas mitigation actions, as an outcome of present-day world-wide negotiations under the UNFCCC umbrella. Section 6 explores possible consequences of the narrowing down of climate sensitivity estimates for the long-term Earth system sensitivity, taking into account 'slow' feedbacks due to the cryosphere response (permafrost melting and ice-sheet disintegration) to a warming world. Implications for the feasibility of avoiding 2 degrees

Celsius of global warming, as required by the Paris Agreement, are briefly discussed.

Finally, in Section 7 some conclusions are drawn.

## 3 Climate sensitivity: a paleoclimatic perspective

### 3.1 Introduction

The climate sensitivity S is defined as the equilibrium global surface temperature change ($\Delta Teq$) in response to a specified

unit forcing (F) according to (Hansen et al., 2012):

$$S = \Delta Teq / F \tag{1}$$

This quantity depends on climate feedbacks, making a distinction between the 'fast-feedback' (Charney, 1979) sensitivity

related to fast hydrological (basically water vapor, cloud and sea ice) responses and the 'long-term equilibrium' sensitivity, related to slow surface-albedo feedbacks (mainly governed by ice-sheet dynamics and vegetation change). The current model-based "best estimate" for the Charney sensitivity amounts to 3 degrees per carbon doubling to 556 ppm, or ¾°C per W/m2, which seems to be confirmed by paleo-climatic observations (Hansen et al., 2008). However, these same observations show the long-term equilibrium sensitivity to even double this amount to 6 degrees per carbon doubling

(equivalent to a nearly ice-free world), an effect not accounted for by the present generation of climate models. Figure 1 (top panel) shows paleoclimatic reconstructions of CO2, CH4 and sea level for the last 425 ky (late Pleistocene), with the rapid





terminations of the four ice ages taking place in this era indicated (from past to present) as T-IV, T-III, T-II and finally T-I, the transition from the Last Glacial Maximum (LGM) to the Holocene at approx. 20 ky till present.

The bottom panel of the figure shows calculated temperatures, based on a fast-feedback sensitivity of ¾°C per W/m2 and doubled forcing because of the slow surface-albedo feedback (middle panel), in good agreement with observations.

**Figure 1.** Paleoclimatic reconstructions of **a)** CO2, CH4 and Sea Level, **b)** Climate Forcing and **c)** Temperature Change for the last 425 ky (late Pleistocene), with the rapid terminations of the four ice ages taking place in this era indicated (from past to present) as T-IV, T-III, T-II and T-I, adapted from Hansen et al., 2008.

30    To relate these results from the late Pleistocene epoch to contemporary climate change, especially the sensitivity and response of the climate system to the rapid, unprecedented emissions of long-lived greenhouse gases during the past century is of great importance. The last time Earth has witnessed such a transition comparable in size but certainly not with respect to pace was Termination-I (T-I), covering a period of approx. 20 ky from the last ice age till present. Also, the last time in Earth



history CO2 level was comparable to the present level of 400 ppm was in the mid Pliocene, approx. 3 My ago. Therefore, in addition to complex climate models simulating contemporary climate change, both periods T-I and mid Pliocene might tell us something more about the long-term effects of the ongoing climate forcing in the present era, the Holocene. In the following both eras will be considered from a climate-feedback perspective. For this purpose the equilibrium global surface

temperature change (ΔTeq) is written as (see, for instance, Hansen et al., 2008):

$$\Delta Teq \quad = f \, \Delta To \qquad\qquad (2)$$
$$= \Delta To + \Delta Tfeedbacks$$
$$= \Delta To + \Delta T1 + \Delta T2 + \ldots,$$

where ΔTo is the global surface temperature change in the absence of climate feedbacks (radiative blackbody damping only), f is the net feedback factor and the ΔTi are increments due to specific feedbacks. As an alternative to the feedback factor f, the gain g may be used to describe the role of climate-feedback processes, with f related to g by:

$f = 1/(1 – g)$ $\qquad\qquad (3)$

Unlike f, a very useful characteristic of the gain g is its additive nature with respect to the individual feedbacks:
$g = g1 + g2 + \ldots$

### 3.2 Feedback analysis: LGM 20ky – Holocene climate change

During the transition from the Last Glacial Maximum to the Holocene, in addition to the fast-feedback Charney sensitivity, "slow" surface-albedo feedbacks were acting upon the climate system because of the retreating ice-sheets and corresponding vegetation change. As described in Hansen, 2008, this powerful feedback was estimated to double the effect of the initial fast feedbacks. Furthermore, CO2 forcing was estimated to be amplified by one-third because of non-CO2 GHGs, such as methane, acting as a (temperature-dependent) feedback in a warming world (Beerling et al. 2009, 2011).

Figure 2a shows the combined amplifying effect of the different feedbacks in the form of an 'equilibrium transfer scheme'.
As a first step, the effect of the methane feedback is incorporated as an 'input amplifier' of ΔTc, the global surface temperature change caused by CO2 forcing only, to ΔTfm. The gain of ¼ in Fig. 2a corresponds to a feedback factor of 4/3, accounting for the increase of one-third of CO2 forcing as reported by Hansen et al. The second part of the transfer scheme describes the amplification by fast (hydrological) feedbacks, from ΔTfm to ΔTf. The fast-feedback gain of 2/3 yields a

feedback factor of 3, in agreement with the current estimate of the Charney sensitivity of 3 degrees per carbon doubling or ¾°C per W/m2. Finally, the effects of ice-sheet retreat are incorporated as an additional (surface-albedo) feedback from ΔTeq to ΔTfm. Together with the fast-feedback gain of 2/3, the additional gain of 1/6 is sufficient to double the feedback



factor from 3 to 6. Thus, given an input amplification of 4/3, the overall climate sensitivity for a specified amount of CO2 becomes 4/3 x 6 = 8°C for 2×CO2, or 2°C per W/m2, in agreement with Hansen et al., 2012.

**Figure 2.** Equilibrium transfer scheme for the transition from the Last Glacial Maximum to the Holocene era, with the methane effect modeled as **a)** an additional input amplifier and **b)** an additional fast feedback from temperature to input.

In Fig. 2b the methane 'input amplifier' is modeled as an additional fast feedback from temperature to input, more in agreement with the physical reality that it is temperature that causes the methane concentration in the atmosphere to increase, and not so much CO2 per se. The gain of 1/24 is scaled in a way that the overall climate sensitivity of 2°C per W/m2 is maintained. Note that this value differs from the gain 1/12 which would result from incorporating the methane influence as an additional fast feedback, increasing this value from 3 to 4, and subsequently multiplying this value by 2 to incorporate the 'doubling effect' of slow feedbacks as was done in the Hansen paper. However, according to the feedback scheme in Fig. 2b, with the slow feedbacks incorporated as an additional gain of 1/6, this would result in an overall sensitivity of 1/(1-(2/3+1/12+1/6)) = 12°C for 2×CO2, not in agreement with the required overall scaling relation of 8°C for 2×CO2, or 2°C per W/m2 as was reported by Hansen et al.





### 3.3 Feedback analysis: Holocene – Pliocene 3My climate

With regard to the slow feedbacks (ice-albedo and vegetation), in Hansen et al. (2012) it is argued that the 'doubling effect' on climate sensitivity during the late Pleistocene epoch from a Holocene perspective is only valid for a negative forcing, because present climate is near the warm extreme of the Pleistocene range. Given the present $CO_2$ concentration of 400 ppm

5    and the corresponding movement toward a warmer climate, a comparison with the mid Pliocene (3 My BP), the last time in Earth history $CO_2$ level was comparable to the present value, might be more relevant. As estimated by Lunt et al. 2010, climate sensitivity during that period was increased by a factor of 1.3-1.5 by the slow surface-albedo feedbacks, instead of the doubling found for the late Pleistocene. The resulting equilibrium transfer scheme is shown in Fig. 3a, where the surface-albedo feedback is incorporated as an 'output amplifier' with a gain of 1/3, corresponding to a feedback factor 1.5. In Fig. 3b

10   this is translated to an overall feedback of 1/9 from output to input, to enable direct comparison with the previous analysis for the LGM-Holocene climate transition.

**(a)**

**(b)**

30

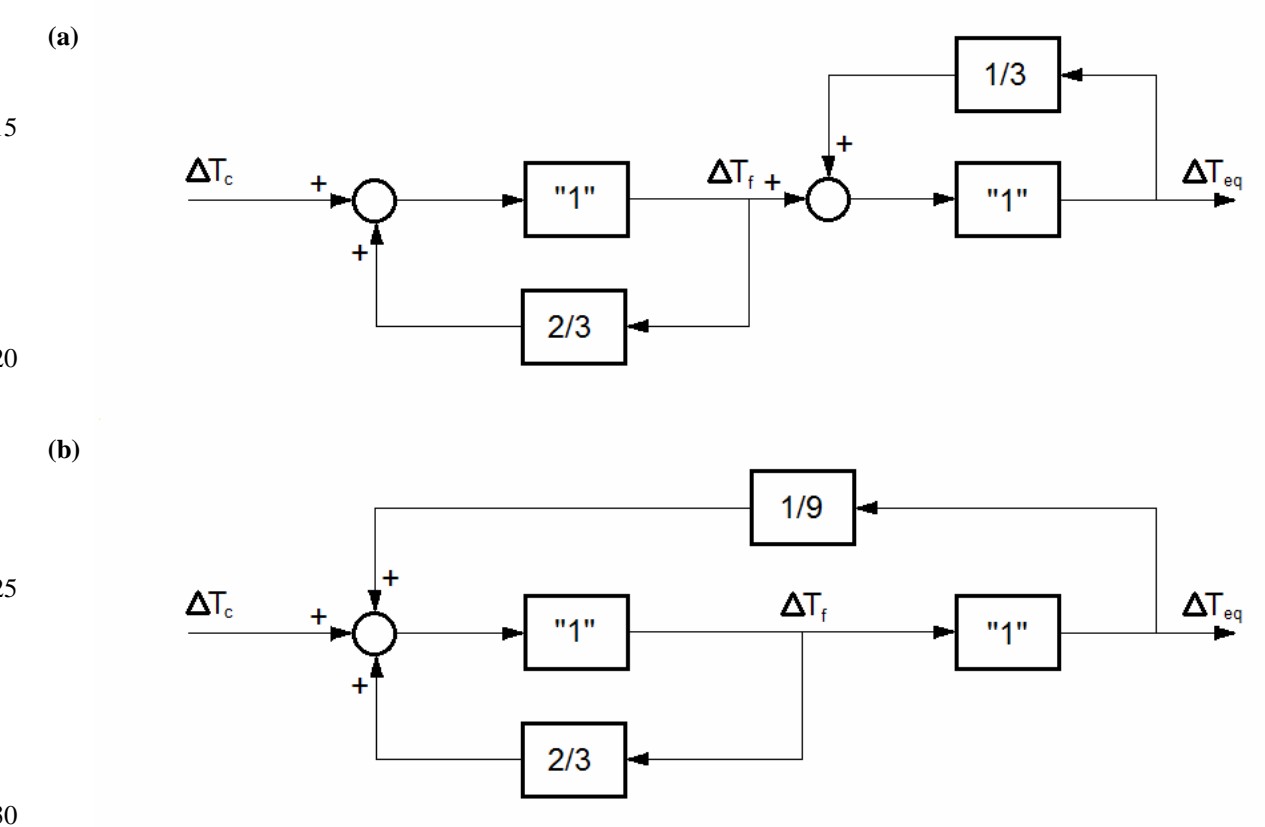

**Figure 3.** Equilibrium transfer scheme for the Mid-Pliocene epoch (3 My before present) with the slow surface-albedo feedback modeled as **a)** an additional output amplifier and **b)** an additional feedback from output to input.



### 3.4 Joint analysis: Contemporary climate change

The key quantity (and uncertainty) at the basis of simulating contemporary climate change towards the end of the current century is the fast-feedback (Charney) sensitivity, as described in Section 3.1. To obtain an indication for climate sensitivity beyond 2100, the paleoclimatic analysis of "slow" feedbacks in Sections 3.2 and 3.3 may be combined to a joint transfer scheme, presented in Fig. 4a. In this figure, the results on methane and surface-albedo feedbacks are simply added, as a first approximation of a combined response.

**Figure 4.** Joint equilibrium transfer scheme for contemporary climate sensitivity, **a)** combining the results of Figure 2b and Figure 3b on fast and slow climate feedbacks from output to input and **b)** making a separation between climate feedbacks incorporated in contemporary climate models ("AR5") and 'amplifiers' beyond the current century, based on paleo-climatic reconstructions ("G3M").





In Fig. 4b, the feedbacks are rearranged in a manner that a separation can be made between the climate sensitivity in 'state-of-the-art' climate models, such as used in the IPCC 5th Assessment Report of 2013 ("AR5"), and the amplifying effects of additional feedback processes on the resulting equilibrium temperature change, based on the paleoclimatic reconstructions from the last Glacial transition and the 3 My BP mid-Pliocene period ("G3M").

Adding the individual gains in Fig. 4b yields an overall gain g of $2/3 + 1/24 + 1/9 = 59/72$, amplifying the fast-feedback factor $f = 1/(1-g)$ of 3, as reproduced by the current climate models, by a factor of approx. 1.8. This is somewhat less than the 'Hansen value' of 2, which would result from a pure serial transfer scheme from input to output: $(4/3 \times 3 \times 1.5)/3 = 2$. More important however, is the sensitivity of f for variations in g. Given the present range of uncertainty of the fast-feedback factor f of climate models, lying between 1.5 – 4.5, this could give totally different long-term results for the two transfer

schemes described here.

In the following, a simple simulation set-up will be described which can be used to study these effects.

## 4 STAGE 2.0 simulation set-up

To analyze the transient effects of greenhouse-gas emissions on the global surface temperature, in the past a simple, spreadsheet-based tool STAGE ('Simple Transfer Analysis of Greenhouse Emissions') has been constructed by the author,

purely for personal clarifying purposes. The underlying dynamics of STAGE is based on a first-order transfer model (see Appendix A), incorporating ocean heat capacity (two levels: surface and deep ocean) and radiative damping to the atmosphere. The strength of the (mainly hydrological) fast feedbacks could be varied by specifying the equilibrium temperature for CO2 doubling (dT2x). The effect of direct (chaotic) atmospheric forcing of the ocean by weather systems was incorporated as stochastic, white noise which was 'reddened' by the slow ocean response (passive regime). Thus, both

the steady state response to CO2 forcing and multidecadal 'unforced' variations of surface temperature could be reproduced. The model output presented in Fig. 5 concerns a 'CO2 doubling' experiment in a forcing pace comparable to actual emissions going on at present and climate sensitivity set to 3°C. Both the equilibrium values dTc (without climate feedback) and dTf (including fast hydrological feedbacks), as well as the transient responses dTs (surface) and dTo (deep ocean) of the partly mixed ocean are shown. For comparison to different emission pathways under study by the IPCC, at the right part of

the graph steady state temperature values of two so-called 'SRES' scenarios (IPCC, 2000) are presented: 'B1', more or less equivalent to the CO2 doubling experiment as described here, and 'A1F', the 'business as usual' scenario with atmospheric CO2 concentration in the year 2100 near to quadrupling since the beginning of the industrial revolution. The two marker values presented in Fig. 5 correspond to the steady state temperature response without climate feedback, to enable direct comparison with dTc.

To extend the model to 'STAGE 2.0', incorporating the 'slow' effects of methane and surface-albedo feedbacks as described in the previous sections, a long-term equilibrium response dTe is added, relating back to $\Delta Teq$ in the joint transfer scheme of Fig. 4b. In Fig. 5 this is shown as a white signal, which amplifies the 3°C fast-feedback response for this case by approx. 1.8,



resulting in an equilibrium temperature change of 5.4°C. Results for other experiments may be directly obtained in a 'what-if' like manner, by either changing the forcings or varying the (fast-feedback) climate sensitivity. In the next section this will be demonstrated in more detail, taking the present-day climate state of our planet as a starting point.

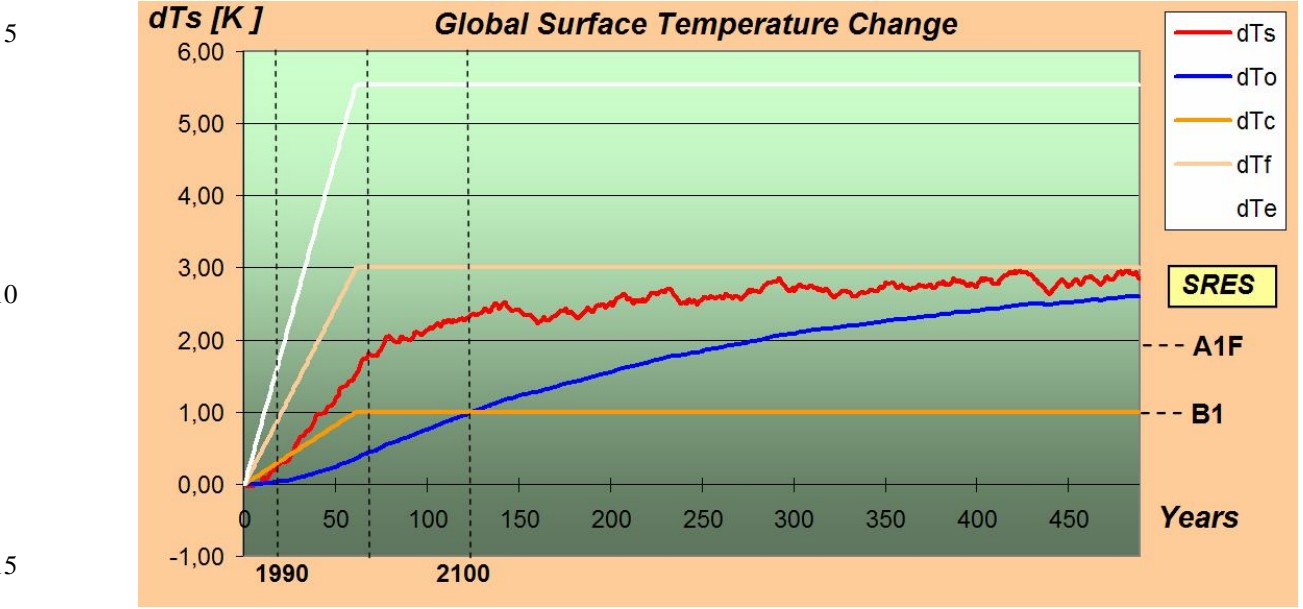

**Figure 5.** STAGE 2 simulation results for the transient analysis of greenhouse-gas emissions. Model output is presented for a fast-feedback climate sensitivity of 3°C for 2×CO2 (equivalent to an atmospheric CO2 concentration of 556 ppm). The long-term equilibrium response dTe is added as a white signal, corresponding to an equilibrium temperature change of 5.4°C.

## 5 Application example: Present-day climate stabilization

At present, since the beginning of the industrial revolution mankind has injected approx. 400 gigatonnes of Carbon into the atmosphere, increasing the more or less stable CO2 concentration over the last 10.000 years of the Holocene from 280 ppm to 400 ppm. The resulting temperature change currently amounts to approx. 1°C, which can be considered as a transient climate response towards a 'short-term' equilibrium if emissions were instantaneously cut to zero. An additional warming of ~0.6°C is estimated to be in the pipeline because of ocean thermal inertia. The situation is summarized in Fig. 6, presenting the STAGE 2 simulation output for a CO2 concentration immediately stabilized at the current value of 400 ppm.

Besides the warming in the pipeline of ~0.6°C because of ocean thermal inertia (In Fig. 6 the gap between dTs and dTf at the time of CO2 stabilization at 400 ppm), simulation results show an additional warming commitment of ~1.4°C (the remaining gap between dTf and dTe), because of long-term feedbacks mainly due to ice-sheet disintegration as described in Section 3.

Overall, in the new equilibrium state global surface temperature is raised approx. 3°C, which is more or less in agreement with paleo reconstructions of mid-Pliocene conditions at 3 My BP (CO2 concentration approx. 400 ppm; global average



temperature 2–3°C higher than pre-industrial, sufficient to raise global sea level by 6–20 m. because of thermal expansion of ocean water and accompanying melt of ice sheets).

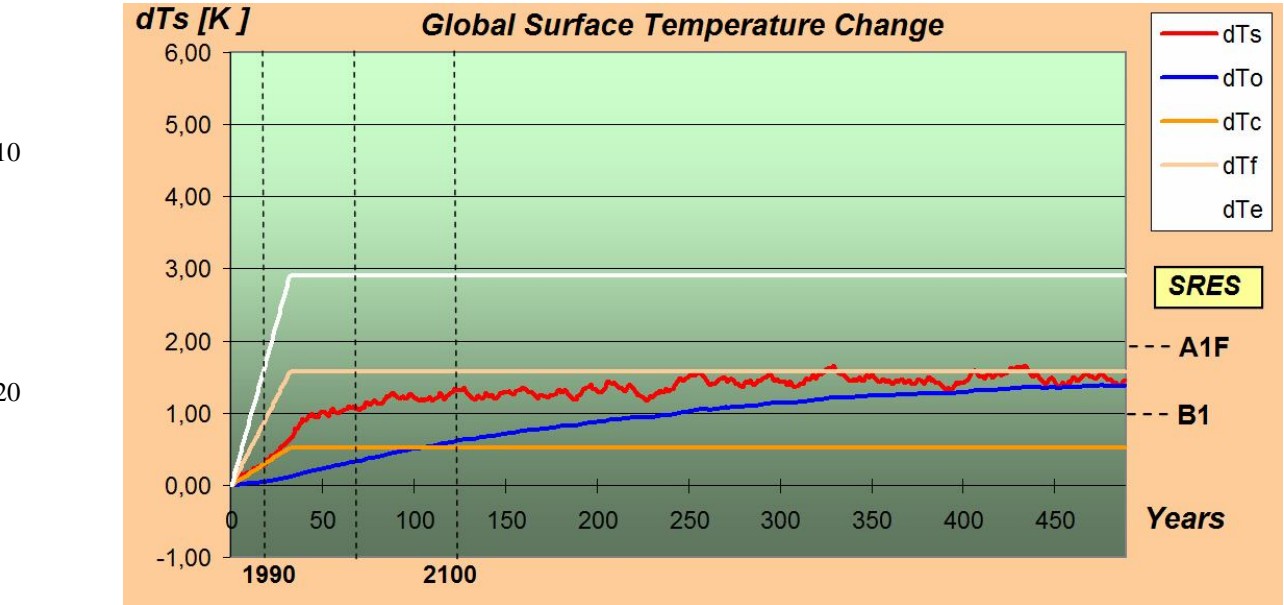

**Figure 6.** STAGE 2 simulation results for a CO2 concentration immediately stabilized at the current value of 400 ppm for a fast-feedback climate sensitivity of 3°C.

The current worldwide negotiations under the UNFCCC umbrella last year have resulted in the so-called "Paris agreement" to stabilize global average temperature to below 2°C above pre-industrial levels and to pursue efforts to limit the temperature increase to 1.5°C. Staying within the 2°C threshold would imply to stabilize the CO2 concentration below 450 ppm, which requires the current fossil-fuel energy consumption to be halted within the next few decades to come.

In Fig. 7a the STAGE 2 simulation output is shown, aiming at the 2°C target described here. In the figure, besides a 2°C threshold line, a 'T-I marker line' has been added, relating to the 5°C temperature change world has witnessed during Termination-I, the dramatic climate transition from the last ice age to present.

Obviously, the 2°C threshold used by the current climate models to determine the required CO2 target to achieve this, tell us only 'part of the deal'. The 'unmodeled' summed-up long-term effects, although relatively small in feedback gain (Fig. 4b), indicate a new equilibrium well above the mid-Pliocene 3My conditions, approaching a 4°C warming world.

Hansen et al. (2008), emphasizing the importance of the long-term feedbacks on climate equilibrium, uses a somewhat different approach to determine a target value for atmospheric CO2. Realizing that these second-order feedbacks are already kicking in, a.o. in the form of present (accelerating) ice-sheet disintegration, instead of relying too much on the models he proposes to stabilize Earth's climate by restoring the planet's measured energy imbalance (currently approx. 0.75 W/m2) as





soon as possible. As a first step this requires a CO2 target concentration of at most 350 ppm, actually a reduction of 50 ppm from the present value of 400 ppm, to be acquired preferably before the end of the current century.

**(a)**

**(b)**

**Figure 7.** STAGE 2 simulation results for **a)** CO2 concentration stabilized at approx. 450 ppm, aiming at a 2 °C threshold for global surface temperature change in agreement with the UNFCCC framework and **b)** stabilizing CO2 at 350 ppm, the initial target recommended by Hansen et al. (2008), aiming at restoring Earth's energy balance.




In the STAGE 2 set-up (Fig. 7b), the effect on climate equilibrium is simulated by stabilizing the CO2 concentration at 350 ppm, roughly the value obtained round 1990.

The simulation results show a 'fast-feedback' stabilization at the present actual value of 1 °C, after which an equilibrium value of approx. 2°C is obtained. However, the strong non-linear amplifying effect of the second-order feedbacks kicking in

5 may be illustrated by increasing the fast-feedback climate sensitivity from 3°C to 4.5°C for 2×CO2, corresponding to the high end of the current 'likely range' of 1.5–4.5 °C (Fig. 8).

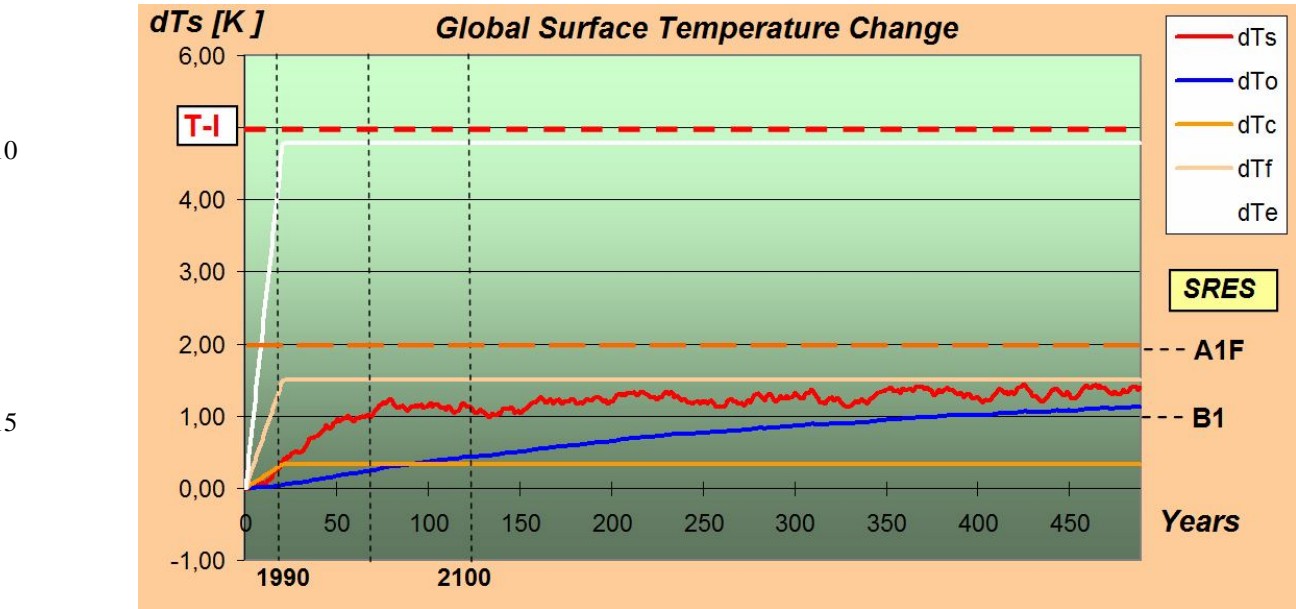

**Figure 8.** STAGE 2 simulation results: amplifying effect of second-order feedbacks for a fast-feedback climate sensitivity of 4.5°C for 2×CO2, instead of the 3°C in Figure 7.

As suggested by Hansen, after obtaining the CO2 target of 350 ppm towards the end of the current century, a further reduction may be needed to minimize the highly non-linear second-order feedback effects as demonstrated here as much as possible. As a final illustration, in Fig. 9 it is examined how much further CO2 reduction it would eventually take to

25 maintain the new long-term climate equilibrium at the current value of +1°C above pre-industrial. The last time this equilibrium value occurred was during the Eemian period approx. 125 ky BP, the last interglacial before the present Holocene, with global sea level 6-9 m. higher than today.





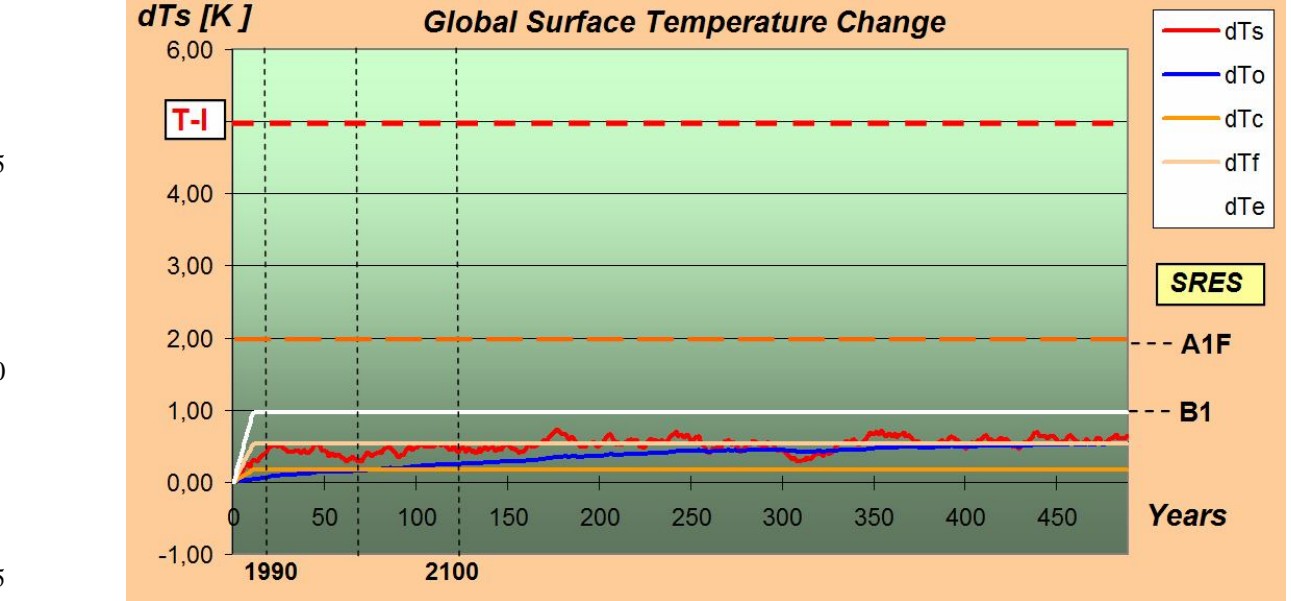

**Figure 9.** STAGE 2 simulation results: maintaining the new long-term climate equilibrium at the current value of +1°C above pre-industrial. The corresponding required CO2 concentration serving as a model input amounts to 314 ppm.

## 6 Narrowing down Equilibrium climate sensitivity: implications for long-term Earth system sensitivity

### 6.1 Introduction

Recently, estimates of the fast-feedback (Charney) climate sensitivity have been 'narrowed down' by constraining climate models by their ability to simulate observed variations in climate (Nature, Cox et al., 2018). It was concluded that the Charney sensitivity, in their study referred to as the 'Equilibrium climate sensitivity' (ECS), has a central estimate of 2.8 °C for 2×CO2, and a 'likely range' of 2.2–3.4 °C which sits towards the middle to lower end of the current estimate of 1.5–4.5 °C. The present section aims at exploring possible consequences of this narrowing down of ECS for the long-term Earth system sensitivity (ESS), taking into account 'slow' feedbacks due to the cryosphere response (permafrost melting and ice-sheet disintegration) to a warming world. The rationale behind this is that signs are present of these 'slow' feedbacks already kicking in, potentially leading to a substantial increase of Earth's temperature response to the radiative forcing caused by ongoing human greenhouse-gas emissions. Hence, the fact that these effects are not accounted for by the present generation of climate models (such as the ones used in the recent Cox et al. study), which nevertheless stand at the basis of international policy making to achieve the '2 degrees Paris Agreement', may be regarded as rather peculiar, if not worrisome.

In the following, without the need for explicitly modeling these (second-order) cryosphere effects as mentioned here, it is shown that already something may be learned from examining the structural relationship between the different feedbacks




involved. Based on this (sensitivity) approach, principal scaling relations between variations in ECS and variations in ESS may be determined.

### 6.2 Method

In their search of an emergent constraint on ECS, Cox et al. (2018) use the simple Hasselmann model (Hasselmann, 1976),
5   relating the variation in global mean temperature $\Delta T$ in response to a change in radiative forcing $\Delta Q$ (see Appendix A for a transfer-function description of this model). The principle parameter of interest to their analysis is the 'radiative damping coefficient' $\lambda$, in their study called the 'climate feedback factor', which with respect to terminology used is somewhat confusing in relation to the feedback factor f and gain g as introduced in Section 3. In Appendix A a relation is derived between this damping coefficient $\lambda$ and the (climate feedback) gain g:

$$g = 1 - \lambda / \lambda_0 \approx 1 - \lambda/4 \quad \text{or, conversely,} \qquad \lambda = \lambda_0 . (1 - g) \tag{4}$$

In this equation $\lambda_0$ is the radiative damping coefficient in the absence of climate feedbacks (radiative blackbody damping only), amounting to 4 W/m2/K. In terms of climate sensitivity an alternative expression for g is given by:

$$g = 1 - 1/\Delta T_{2x}.4/\lambda_0 \approx 1 - 1/\Delta T_{2x} \tag{5}$$

with $\Delta T_{2x}$ the equilibrium surface temperature change for a doubling of CO2 forcing, the Equilibrium climate sensitivity ECS.
20   This yields the following 'scaling scheme' from ECS to ESS:

| 1. | for a given climate sensitivity $\Delta T_{2x}$ the fast-feedback contribution to the gain g is calculated according to Eq. (5) |
|----|----|
| 2. | from this the overall gain g is calculated by adding the fast-feedback and long-term equilibrium components: g = g1 + g2 |
| 3. | the total earth-system feedback factor f is derived from g according to Eq. (3) |

which results in the Earth system sensitivity ESS.



An alternative way to achieve this scaling is based on a direct, analytical expression for the total earth-system feedback factor f (see, for instance, Buchdahl, 1999):

$$f = f1.f2 \, /(f1 + f2 - f1.f2) \ = \ \Delta T2x \, .f2 \, /(\Delta T2x + f2 - \Delta T2x \, .f2) \tag{6}$$

with f2 the combined feedback factor of the 'slow' second-order feedback processes.

In terms of ESS and ECS this yields the following (non-linear) scaling relation:

$$ESS = \ ECS.f2 \, /(ECS + f2 - ECS.f2) \tag{7}$$

In Appendix B an estimate of f2 is derived from paleoclimatic observations, showing the fast-feedback sensitivity on the long-term possibly to be doubled by the cryosphere response. This corresponds to the following linear scaling relation, from now on referred to as the 'Series model':

15 $$ESS = \ 2.ECS \tag{8a}$$

In the appendix it is shown that for an ECS of 3 an additional gain of 1/6 is already sufficient to achieve this doubling, corresponding to a feedback factor f of 6/5 (Eq. 3). For arbitrary ECS, substituting this value for f2 in Eq. 7 after reduction to lowest terms yields the following non-linear scaling relation between ESS and ECS (from now on called the 'Cascade' 20 model):

$$ESS = \ 6.ECS \, / \, (6 - ECS) \tag{8b}$$

Note that for the 'classical case' of ECS = 3 both scaling relations yield an ESS of 6, doubling the original fast-feedback 25 sensitivity in agreement with the paleoclimatic observations. In the following, both scaling models will be contrasted to each other, to demonstrate the sensitivity of long-term effects to short-term 'deviations' from this default value. Of specific interest are the new constraints put on ECS by Cox et al. (2018) by analyzing the ability of coupled ocean-atmosphere climate models in the CMIP5 archive to simulate observed variations in climate.

### 6.3 Results

30 As a conclusion of the Cox et al. (2018) study a new a 'best estimate' for ECS of 2.8 °C has been formulated, with a 'likely' range with 66% confidence limits of 2.2–3.4 °C. This 'likely' range sits towards the middle to lower end of the Intergovernmental Panel on Climate Change (IPCC) estimate of 1.5–4.5 °C (with a central value 3°C), which has remained



unchanged for more than 25 years. However, in this period several initiatives have been undertaken to adjust this value, leading to different alternative estimates either to the low range, for instance 1.2–3.0 degrees with a 'best estimate' of 1.7 °C as formulated by Lewis, or the high range of 2.7–4.5 degrees Celsius, with a 'best estimate' of 3.4 °C as formulated by Fasullo (Strengers, 2014). Below this 'history' of ECS estimates until the most recent value is summarized (Table 1), and

5 extended to ESS according to the scaling relations of Eq. 8a,b (Fig. 10).

**Table 1.** *Best estimates and likely ranges* (i.e. 66% probability) of the Equilibrium climate sensitivity (ECS) and Earth system sensitivity (ESS). ESS is obtained from ECS by a scaling factor (between brackets) according to either a Series or Cascade scheme as derived in Section 6.2.

| ECS (°C) | *from* | *Best estimate* | *to* |
|---|---|---|---|
| 'High range' | 2.7 | 3.4 | 4.5 |
| IPCC AR5 'likely range' | 1.5 | 3.0 | 4.5 |
| Cox et al. 'likely range' | 2.2 | 2.8 | 3.4 |
| ESS (°C) | | | |
| Series model | 4.4 (2.0) | 5.6 (2.0) | 6.8 (2.0) |
| Cascade model | 3.5 (1.6) | 5.2 (1.9) | 7.9 (2.3) |

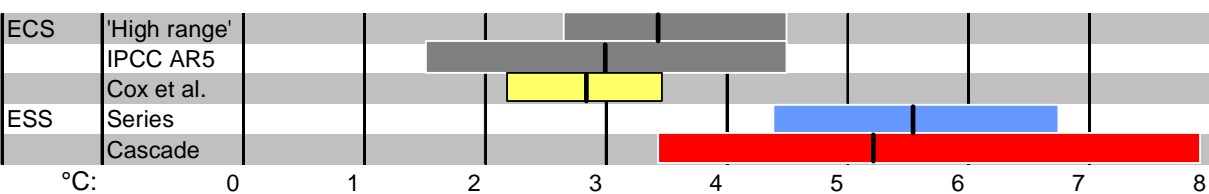

15 **Figure 10.** Bar chart presentation of '*Best estimates' and 'likely ranges'* (i.e. 66% probability) of ECS and ESS. The yellow bar refers to the new ECS estimate by Cox et al. (2018).

In Fig. 11a. results are shown for ESS and ECS as a function of the radiative damping coefficient $\lambda$, with a maximum value of 4 W/m2/K, roughly the value in the absence of climate feedbacks (radiative blackbody damping only). For each $\lambda$, a value for the climate sensitivity is given by the theoretical relation ECS $= \Delta Q_{2x} / \lambda$, with $\Delta Q_{2x} \approx 4$W/m2, the radiative forcing

20 for a doubling of atmospheric CO2. To relate this value of ECS to the CMIP5 model archive, the values of $\lambda$ and ECS for the different CMIP models (see Appendix C) are presented in the figure as black dots. The yellow interval between $\lambda$=1 and $\lambda$=1.5 more or less corresponds to the range of CMIP5 models as selected by Cox et al. for their new estimate of the ECS



'likely' range as presented in Table 1 and Fig. 10. Subsequently, 2 values for ESS are presented, scaled according to either the linear 'Series' model (Eq. 8a) or non-linear 'Cascade' model (Eq. 8b). In Fig. 11b, the corresponding scaling factor is presented by normalizing the values of Fig. 11a with respect to ECS.

5 **(a)**

**Climate feedback gain g**

λ (W/m2/K)

**(b)**

**Climate feedback gain g**

λ (W/m2/K)

30 **Figure 11.** ESS and ECS as a function of the radiative damping coefficient λ, **a)** scaled according to either the linear 'Series' model or the non-linear 'Cascade' model (Eq. 8a,b), with the theoretical ECS plotted against 'raw' model data from the CMIP5 model archive (black dots). In the top part of the figure, the radiative damping coefficient λ is mapped onto the theoretical 'climate feedback gain', given by Eq. 4. and **b)** normalized with respect to ECS.



**6.4 Discussion**

As presented in Table 1 and Fig. 10, the 'narrowing down' of the ECS estimate in the recent Cox et al. study on the longer term still may lead to values in a range well beyond the original IPCC and 'high' ranges by the 'scaling up' process from ECS to ESS. This is supported by Fig. 11, showing the rapid (non-linear) increasing role of the cryosphere contribution (red

curve) to the overall Earth system sensitivity ESS for a decreasing radiative damping coefficient $\lambda$.

From a stability perspective, within the range of the total CMIP5 model archive (black dots in the figures) the non-linear scaling process as described here leads to a relatively high sensitivity of the total earth-system response to variations in $\lambda$. The model selection made by Cox et al. restricts this range to values between approx. 1 to 1.5 W/m2/K, shown as the yellow

bar in the bottom part of the figure. For the lower limit of $\lambda \approx 1$ this corresponds to a 'fast-feedback' gain of 0.75, shown in the top part of the figure as the 'Climate feedback gain g'. As described in Appendix B, together with the cryosphere gain of 1/6 this translates in a total gain margin of 1 – (3/4 + 1/6) = 1/12, leaving hardly any room for 'unforeseen feedbacks', by definition not accounted for by the CMIP5 models.

Obviously, using models to determine the required CO2 target to achieve a temperature 'setpoint', like the 2°C threshold according to the Paris Agreement, tells us only 'part of the deal': 'unmodeled' summed-up (long-term) effects, although relatively small in feedback gain may push the total Earth climate system towards a new equilibrium well beyond the initial temperature threshold.

More in line with the stability paradigm, the precautionary approach to determine a target value for atmospheric CO2 by

Hansen et al. (2008) as described in Section 5 seems to offer a better alternative. Emphasizing the importance of 'long-term' feedbacks on climate equilibrium, and realizing that these second-order feedbacks are already kicking in, a.o. in the form of melting permafrost and (accelerating) ice-sheet disintegration, it is proposed to stabilize Earth's climate by restoring the planet's energy balance as soon as possible on the basis of direct observations, instead of relying too much on models such as in the CMIP5 archive.

**7 Conclusion**

A simulation tool (STAGE: 'Sensitivity Transfer Analysis of Greenhouse Emissions') has been constructed with which the sensitivity of the Earth climate system to human interference can be studied. The tool is intended to offer users of the software, e.g. students in higher education, support in gaining insight into critical aspects of the climate system and thus counteracting common misconceptions with regard to the functioning of this system.



The conceptual design of STAGE is based on the paradigm *'learning as experimenting'*, encouraging students to explore climate sensitivity in its various aspects in an active manner.

The main learning objective is to 'get a feel' for both 'short-term' (current century) and possible 'long-term' (beyond) consequences of greenhouse-gas mitigation measures.

5   To achieve this, the classical notions of climate sensitivity and feedback, as used to characterize present-day climate models, were treated from a broader (paleoclimatic) perspective. This served as a basis for extending the simulation core of a previously developed tool for the transient analysis of greenhouse-gas emissions to incorporate long-term effects (STAGE 2.0).

The feasibility of the educational application was investigated by exploring possible long-term climate consequences and
10   uncertainties of near-term greenhouse-gas mitigation actions, as an outcome of present-day world-wide negotiations under the UNFCCC umbrella. It was demonstrated that using models to determine the required $CO_2$ target to achieve a temperature 'setpoint', like the 2°C threshold according to the Paris Agreement, tells us only 'part of the deal': 'unmodeled' summed-up (long-term) effects, although relatively small in feedback gain may push the total Earth climate system towards a new equilibrium well beyond the initial temperature threshold.



**Appendix A: Simple Transfer Analysis**

To simulate and explore the dynamic response of the Earth climate system to variations in radiative forcing, simple models in structure similar to the one proposed by Hasselmann (1976) may be useful. At the core of these models stands a first-order

5  transfer function, linking changes in radiative forcing $\Delta Q$ to the global mean surface temperature change $\Delta Ts$ (Fig. A1).

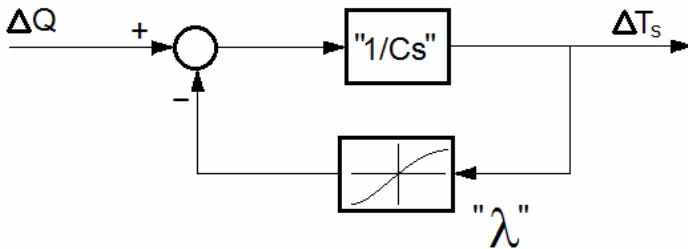

**Figure A1.** Transfer from radiative forcing to global surface temperature change.

15  According to the block diagram the net incoming radiation flux is integrated and stored as additional energy in the system, increasing the outgoing radiation flux until a new thermal equilibrium is obtained. The corresponding transfer function is given by:

$$\Delta Ts/\Delta Q(s) = 1 / (Cs + \lambda) \tag{A1}$$

with C the system's (mainly ocean) heat capacity and $\lambda$ the radiative damping coefficient back to space. This corresponds to

a first-order transfer function with a DC gain of $1/\lambda$ (setting $s = 0$ in equation A1) and time constant $\tau = C / \lambda$.

To incorporate the role of temperature-dependent climate feedbacks, in Fig. A1 the radiative damping coefficient $\lambda$ is

expressed as a non-linear function of $\Delta Ts$. A value of special interest is the radiative damping for a doubling of CO2 forcing

25  since the beginning of the industrial revolution (approx. 4 W/m2), which may be written as:

$$\lambda = 4 / \Delta T2x \tag{A2}$$



with $\Delta T_{2x}$ the equilibrium surface temperature change. In absence of climate feedbacks (radiative blackbody damping only) this value, defined as $\lambda_0$, amounts to 4 W/m2/K, resulting in an equilibrium temperature change $\Delta T_0$ of about 1°C. The equilibrium temperature change $\Delta T_{2x}$ in the presence of feedbacks is defined as the (fast-feedback) climate sensitivity. In terms of the feedback factor f and the (feedback) gain g, as defined in Section 3 and related to each other by f = 1 / (1 – g), or g = 1 – 1 / f, this yields:

$$f \ = \ \Delta T_{2x} / \Delta T_0 \ = \Delta T_{2x} \ . \ \lambda_0/4 \tag{A3}$$

$$g = 1 - 1/\Delta T_{2x} \ .4/\lambda_0 \tag{A4}$$

Finally, combining equations (A2) and (A4) yields a relation between the 'positive feedback' gain g, used in the sensitivity analysis, and the 'negative feedback' gain $\lambda$ in Fig. A1:

$$g = 1 - \lambda \ / \ \lambda_0 \ \approx 1 - \lambda/4 \quad \text{or, conversely,} \qquad \lambda = \lambda_0 \, .(1 - g) \tag{A5}$$

**Appendix B: Sensitivity Analysis**

Before the recent Cox et al. study on ECS the model-based "best estimate" for the Charney sensitivity amounted to 3 degrees per carbon doubling, or ¾°C per W/m2, which seems to be confirmed by paleoclimatic observations (Hansen et al., 2008). However, these same observations show the long-term equilibrium sensitivity to even double this amount to 6 degrees per carbon doubling (eventually resulting in a nearly ice-free world), an effect not accounted for by the present generation of coupled ocean-atmosphere climate models such as the ones in the CMIP5 ensemble used as input to the study. Although in the Hansen paper it is argued that this 'doubling effect' on climate sensitivity is strictly speaking only valid for a negative forcing, and probably less in a warming world, in the following this doubling is maintained as an overall estimate of combined cryosphere feedbacks caused by ice sheet retreat (surface albedo) and melting permafrost (methane).
Fig. B1 shows the combined amplifying effect of these different feedbacks in the form of an 'equilibrium transfer scheme'. The first part of the transfer scheme describes the amplification by fast (hydrological) feedbacks from ΔTc, the global surface temperature change caused by CO2 forcing only, to ΔTf. The fast-feedback gain of 2/3 corresponds to a feedback factor f of 3 (Eq. 3), in agreement with the current estimate of the Charney sensitivity of 3 degrees per carbon doubling. In




the second part the additional doubling effect of slow second-order feedbacks is incorporated as an 'output amplifier' of a factor 2 from ΔTf to ΔTeq, resulting in an overall feedback factor of 6 from ΔTc to ΔTeq.

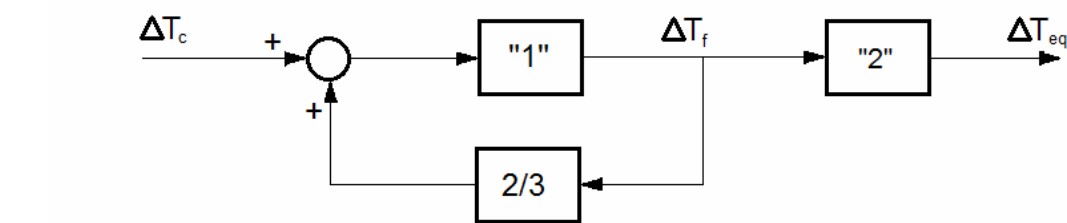

**Figure B1.** Equilibrium transfer scheme for the combined amplifying effects of fast and long-term equilibrium feedbacks. The numbers between quotes refer to the "steady state" values of dynamic quantities.

In Fig. B2a this 'output amplifier' is rearranged as a local feedback amplifier with a gain of 1/2.

**(a)**

**(b)**

**Figure B2.** Rearranging the (Cryosphere) output amplifier as a (local) feedback amplifier **(a)** and as an additional feedback from temperature to input **(b)**.




In Fig. B2b the feedback concept is extended from temperature to input, more in agreement with the physical reality that the additional warming caused by the cryosphere (melting permafrost and retreating ice sheets) will also have impact on the combined ocean-atmosphere temperature response. The gain of 1/6 is scaled in a way that the overall climate sensitivity of 6°C per W/m2 is maintained. In Section 6 this is referred to as the 'Cascade model', as opposed to the 'Series model' of Fig. B2a.

In this figure the fast-feedback gain of 2/3 corresponds to the 'Best estimate' of ECS of 3 °C per CO2 doubling, based on the 'CMIP5' ensemble of coupled ocean-atmosphere models. In absence of this fast-feedback, an overall gain of 1/6 remains, corresponding to a feedback factor of 6/5 (Eq. 3 in Section 3).

For arbitrary ECS, as derived in Section 6, substituting this value for f2 in Eq. 7 after reduction to lowest terms yields the following scaling relation between ESS and ECS (Cascade model):

$$ESS = 6.ECS / (6 - ECS) \tag{B1}$$

For the Series model, according to Fig. B2a, this becomes:

$$ESS = 2.ECS \tag{B2}$$

*Stability considerations*

As can be seen from Eq. B1, for the Cascade model a singularity arises for ECS = 6, leading to an 'infinite' value of ESS. By applying Eq. 5 of Section 6, the corresponding fast-feedback gain is given by: $g = 1 – 1/ECS = 5/6$, replacing the 2/3 in Fig. B2b. Indeed, according to this figure, adding the cryosphere gain of 1/6 yields an overall gain of 5/6 + 1/6 = 1, making the system unstable.

In a manner similar to the application of control theory to the linear stability analysis of dynamical systems, a useful concept in the present sensitivity-analysis context is the so-called 'gain-margin' gm:

$$gm = 1 – \sum_i g_i \tag{B3}$$

which for a given overall gain g provides insight into the additional gain allowed before the stability condition becomes violated. For a fast-feedback gain of 2/3 this becomes:

1 – (2/3 + 1/6) = 1/6, demonstrating the crucial role the cryosphere plays in the overall earth-system stability for the Cascade scheme. For the Series model of Eq. B2 this becomes 1 – 2/3 = 1/3, regardless of the cryosphere contribution, which only serves as a, constant, factor 2 'output' amplifier (Fig. B2a).



## Appendix C: CMIP5 Models

**Table C1.** Earth system models used as input to the Cox et al. (2018) study, as provided by the CMIP5 project.

|     | Model | $\lambda$ (Wm$^{-2}$ K$^{-1}$) | ECS (K) | $Q_{2xCO2}/\sigma_N$ | $\Psi$ (K) |
|-----|-------|---------|---------|---------|---------|
| a   | ACCESS1-0     | 0.8 | 3.8 | 8.5  | 0.22 |
| b   | CanESM2       | 1.0 | 3.7 | 8.3  | 0.17 |
| c   | CCSM4         | 1.2 | 2.9 | 7.3  | 0.19 |
| d   | CNRM-CM5      | 1.1 | 3.3 | 8.7  | 0.16 |
| e   | CSIRO-MK3-6-0 | 0.6 | 4.1 | 6.1  | 0.21 |
| f   | GFDL-ESM2M    | 1.4 | 2.4 | 5.9  | 0.15 |
| g   | HadGEM2-ES    | 0.6 | 4.6 | 7.8  | 0.29 |
| h   | inmcm4        | 1.4 | 2.1 | 11.9 | 0.07 |
| i   | IPSL-CM5B-LR  | 1.0 | 2.6 | 7.2  | 0.16 |
| j   | MIROC-ESM     | 0.9 | 4.7 | 11.7 | 0.23 |
| k   | MPI-ESM-LR    | 1.1 | 3.6 | 11.9 | 0.15 |
| l   | MRI-CGCM3     | 1.2 | 2.6 | 9.3  | 0.09 |
| m   | NorESM1-M     | 1.1 | 2.8 | 7.8  | 0.14 |
| n   | bcc-csm1-1    | 1.1 | 2.8 | 6.9  | 0.19 |
| o   | GISS-E2-R     | 1.8 | 2.1 | 11.1 | 0.12 |
| p   | BNU-ESM       | 1.0 | 4.1 | 8.0  | 0.15 |
| f$^x$ | GFDL-ESM2G  | 1.3 | 2.4 | 7.1  | 0.20 |
| f$^y$ | GFDL-CM3    | 0.8 | 4.0 | 6.7  | 0.36 |
| i$^x$ | IPSL-CM5A-LR| 0.8 | 4.1 | 8.6  | 0.20 |
| j$^x$ | MIROC5      | 1.5 | 2.7 | 10.2 | 0.23 |
| n$^x$ | bcc-csm1-1-m| 1.2 | 2.9 | 7.4  | 0.14 |
| o$^x$ | GISS-E2-H   | 1.7 | 2.3 | 11.8 | 0.10 |



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
