# Peer review of "STAGE 2.0: Sensitivity Transfer Analysis of Greenhouse Emissions"

_Geoscience Communication, 2018_

## Referee Comment (RC1) · D. Crookall (Referee) · 14 Jun 2018

Review by David Crookall, crookall.consulting@gmail.com, 14 June 2018,

Of Peter O. Passenier's article:

STAGE 2.0: Sensitivity Transfer Analysis of Greenhouse Emissions,

Submitted to *Geoscience Communication* – Copernicus

I will write this in a more personal manner than some reviews are done; it helps simplify language. I will also write this as I read the ms.

Your ms, Peter, addresses some important issues and presents a learning tool to communicate aspects of climate change. My impression is that your ms is still in a somewhat embryonic form, and needs some considerable modification and development to be published in *Geoscience Communication*. Indeed, it is a pity that you submitted this early draft 'so soon'. Here are some indicators of what I would consider to be necessary or highly desirable changes.

**Language/grammar**. Shorter and simpler sentences would improve readability and clarity. I urge you to work with a good copy editor once you have your final draft to submit. Please remember that, in much academic publication, language and clarity are greater determinants of being cited than are novelty and strength of content.

**Objectives**. It seems to me that you have two sets of objectives:

1. Objectives of your article, in relation to the potential reader.
2. Objectives of your simulation, for users (students, et al).

Keep these two sets separate. A potential reader needs to know very early on (in the abstract and the intro) if the article is of potential interest. If the article objectives are not clear, you will lose the reader. Here are two suggested sentences: *The main objective of this article is to present a pedagogical tool that takes the form of an interactive simulation for use in university courses on climate change. The purpose of the simulation is twofold: (1) to help students understand the sensitivity of the climate system, especially in regard to GHG emissions, and (2) to provide students with insight into the long-term consequences of global warming.*

**Article objectives**. I found it difficult to figure out what objectives you have for the article. Early on you talk about your simulation and its purpose of helping students understand CC. However, later, in Section 3, you go on to talk about the workings and adequacy of various climate models, without (I think) clearly showing how the intricate nature of these climate models and what seems to be your suggested model

fit into your simulation for students. In my view Section 3 belongs in a journal on climate modelling.

I may be missing something, but I also found it hard to understand how your Sections 4 and 5 relate to your simulation? Section 4 is titled "simulation set-up", which made me expect to see aspects of the practical working of your simulation in the classroom, such as, starting the simulation, interface configuration, student interaction with the simulation, etc. However, I then encountered more discussion about the climate system, and not about your simulation, how it is used, and what results you have obtained in using it. Section 6 seems to do something similar.

Your **Methods** section provides interesting mathematical analysis and model comparison. However, is this the objective of your simulation, and is this what you wish your students to learn? If so, then it does not seem to match (fully?) your objectives as laid out at the start.

All that gives the impression that your accomplished (as opposed to your stated) **objective** is to discuss aspects of climate modelling. Of course, that is of crucial importance, and is discussed in several climate journals, two of which are published by the AGU and the EGU. Again, *Geoscience Communication*, as eclectic and interdisciplinary as it is, does not, in my view stretch that far, unless it is to discuss how such models can communicate aspects of geoscience. May I suggest that you bring out this aspect of your article: How does your tool (better) communicate, and how do we know that it does?

My suggestion is to focus on your simulation as a **learning tool**. For example, you could discuss (and measure and analyse) the various ways in which students benefit from your simulation – how your simulation is revived by, and communicates climate change to, a given audience. You can of course, focus on a (small) selection of (already existing) models, and allow students to manipulate the various variables and feedbacks in the climate system: GHG behaviour, feedback loops (eg, increase in atmospheric H2O, thawing of CH4 substrates), albedo effects, etc. This audience is likely to be a more sophisticated audience.

At one point you say "The main learning objective is to 'get a feel' for both 'short-term' (current century) and possible 'long-term' (beyond) consequences of greenhouse-gas **mitigation** measures." However, I could not find how students in your simulation would manipulate variables representing mitigation measures and thus see their effects.

You say that "The conceptual design of the tool is based on the paradigm '**learning as experimenting**', encouraging students to explore climate sensitivity in its various

aspects in an active manner.".  In my view, this is what your article should focus on, not the detailed analysis of the validity of various climate models.  You state that your tool aims to counter common misconceptions [regarding the climate system].  This sounds like it is intended for a lay or general-public audience, not students specializing in climate models.  If so, then the mathematics will not serve any purpose.  In a simulation, you need to strike a careful balance between detail (a high fidelity simulation) and relatively simple pedagogically-useful simulation.  The beauty of pedagogical simulation is its ability to represent reality at the most useful level of representivity for a given audience (sufficiently simple to provide powerful insight and so as not to swamp the main message, and sufficiently complex to provide realistic insight);  the more sophisticated (educated) the audience, the more complex the simulation.  At a certain point of increasing sophistication, the simulation will cease to have a pedagogical purpose and will manifest a research purpose, poorly adapted to helping lay people understand.  Some useful discussion on this has been published, and could be cited.

You mention "the paradigm '**learning as experimenting**', encouraging students to explore climate sensitivity in its various aspects in an active manner".  You really do need to provide references to this method.  I would probably not call it a paradigm, but rather a method or an approach.  A widely used approach goes under the name of the '**experiential learning cycle**', pioneered by Dave Kolb.  Simulation designed and conducted within the framework of this cycle tends to provide a powerful method for communication, and is thus relevant to *Geoscience Communication*.  However, a word of warning:  No game, simulation or role-play or similar activity can hope to produce its full learning potential without the crucial step of **debriefing**.  For the purpose of debriefing, it would be useful for facilitators and participants to have the underlying model, probably in non-mathematical terms, for example, as a system dynamics model.  One question in the debriefing might cover 'what if' this or that variable in the model were modified?

I am not sure if you have done this, but a study on the **communicative effectiveness** for the various audiences of your simulation would contribute greatly the literature.  Simulation can be a particularly powerful tool for communicating insights into geoscience phenomena, but it and its use (including the debriefing) need to be evaluated in a rigorous manner.

So, Peter, I hope these notes will help you to reshape your ms, to provide more structure, to decide on and achieve clear objectives, and to keep in mind the purpose of *Geoscience Communication*.  The Edito by Illingworth et al is especially useful.

---

## Referee Comment (RC2) · Anonymous Referee #2 · 28 Aug 2018

This article presents the conceptual and technical background to a simulation tool designed to explore the sensitivity of the Earth climate system to human interference. Communicating and conveying the critical thresholds and feedbacks within the climate systems to non-specialists is a laudable objective, and the aim of the study is to 'describe the realization of an educational simulation tool' ...'which offers students the possibility to explore climate sensitivity'. The set-up therefore is the exploration of a pedagogic tool, based around the concept design of 'learning as experimenting'.

The difficulty is that the educational framework for the study and the pedagogic context that it is applied to are not discussed - the simulation tool is vaguely described as being for 'students in higher education'. There is no discussion of the level of these HE students (introductory vs advanced, general vs specialist), how the tool is expected to be

used by students and educators, and what level of prior knowledge or understanding of climate science is expected from them. Much of the paper is at a fairly advanced technical level, and it is doubtful that anyone without a reasonable working knowledge of climate models would follow its thread (and I'm fairly sure would be beyond the general science-informed reader). Moreover, there is no discussion of other climate simulation tools that have been used in educational settings, or of educational studies that have pursued this learning as experimenting approach. The abstract mentions common misconceptions but what are these? - they ought to be far more clearly signposted in the text, being implicit in the discussion rather than explicitly highlighted.

Thus, despite being pitched as an educational tool, the educational basis of this study is not developed. To be of use to the educational community, one would expect to see some degree of evaluation of the effectiveness of this simulation tool in improving the understanding of 'students' in climate sensitivity. There does not seem to be any indication that this simulation has actually been trialled and tested on the intended audience.

In summary, the manuscript does a poor job of explaining who it is for, what its objectives are and what its key messages are. As noted above, this is essentially the technical outline of a simulation tool, with much of the substance relating to explaining how sensitivity was defined and operationalised within the software. In that regards, as an exemplar of an innovative numerical simulation, this paper would perhaps be better suited to a technical 'geoscience and computers' journal. But in terms of helping to communicate climate sensitivity to non-specialists, far more attention needs to be given to demonstrating its pedagogic rigor and practical efficacy. To do that would require a fundamental re-focusing and re-organisation of the study, which is too large a task to be major revision, hence my recommendation to reject.

---

## Author Comment (AC1) · 3 Sep 2018

Thank you very much for your very useful and constructive comments to my manuscript.

Although you recognize the potential importance of the issues addressed in the study, considerable modifications and development are foreseen to make it fit for publication in Geoscience Communication. An important factor in this is to have a clear set of objectives for both readers of the article and possible (end)users the work is intended for. I agree that on both parts a lot has to be gained in the manuscript, thus avoiding possible confusion or (even worse) total loss of interest during the reading process. Hence, before specifically responding to your very valuable suggestions for improve-

ment (provided in the supplement of this comment), it's maybe better to first provide some short background regarding my thoughts on the what's and the why's of the research reported here.

The basic idea was to develop a simple, but multi-scale, simulation model (in the paper perhaps somewhat confusingly referred to as the 'simulation tool'), which because of its simplicity is suitable for 'sensitivity experiments' (basically 'what-if' analyses) in the field of (university-level) climate-change education. Because of this multi-scale aspect, possible long-term cryosphere influences on climate sensitivity may be assessed, albeit in a more qualitative way, which nowadays already seem to become more and more dominant in the Earth system response to GHG forcings. To achieve this, the 'different worlds' of paleoclimatic observations and contemporary (complex) climate models had to be united, which is what the different sections of the manuscript are about. So, indeed, the study at first sight seems to focus on climate-model scaling and validation issues, but as a result actually a multi-scale simulation core is realized which because of its simplicity is directly suitable to be integrated in an educational setting.

Hence, in terms of objectives of the simulation, typical 'users' are courseware (e.g. TEL, CAI, MOOC, etc.) developers in the field of climate-change education, which can enrich their available climate-model software libraries with an innovative module geared towards a more effective communication of the complexity of the functioning of the Earth climate system to the 'student'.

In terms of potential readers of the article, besides the aforementioned educational software developers, the 'public horizon' may be extended to other actors (teachers, (post)graduate students, etc.) in some way involved in the climate-change curriculum development process. My basic assumption was that these 'different types' of potential readers all might appreciate the goals and topics covered by the new Geoscience Communication Journal, as laid down in the editorial by Illingworth et al. Thus, by increasing public outreach this way, my hope is to raise potential interest in the climate-change educational community ('Building bridges'), possibly to a level were the model

presented here may be integrated and tested with end users in an actual educational setting.

Please find my more detailed response to the necessary or highly desirable changes as proposed by you in the supplement to this comment.

Please also note the supplement to this comment:
https://www.geosci-commun-discuss.net/gc-2018-5/gc-2018-5-AC1-supplement.pdf
* * *
[Figure]

**Supplement:**

I will now address the necessary or highly desirable changes as proposed by you.

**Language/grammar**.
"Shorter and simpler sentences would improve readability and clarity. I urge you to work with a good copy editor once you have your final draft to submit. Please remember that, in much academic publication, language and clarity are greater determinants of being cited than are novelty and strength of content."

I agree that both readability and clarity could benefit from a good copy editor, necessary actions with respect to professional proofreading are foreseen in case of final draft submission.

**Objectives**.
"It seems to me that you have two sets of objectives:

> 1. Objectives of your article, in relation to the potential reader.
> 2. Objectives of your simulation, for users (students, et al).

Keep these two sets separate. A potential reader needs to know very early on (in the abstract and the intro) if the article is of potential interest. If the article objectives are not clear, you will lose the reader. Here are two suggested sentences: *The main objective of this article is to present a pedagogical tool that takes the form of an interactive simulation for use in university courses on climate change. The purpose of the simulation is twofold: (1) to help students understand the sensitivity of the climate system, especially in regard to GHG emissions, and (2) to provide students with insight into the long-term consequences of global warming.*"

This is probably by far the most crucial mission in the revision of my manuscript, given the basic considerations in the above presented background thoughts about my article. I have (re)formulated the different objectives accordingly in a concise manner (in line with your two suggested sentences) as follows:

*The main objective of this article is to present a simulation model that may be integrated in a pedagogical tool for use in university courses on climate change. The purpose of the simulation is twofold: (1) to help students understand the sensitivity of the climate system, especially in regard to GHG emissions, and (2) to provide students with insight into the long-term consequences of global warming.*

Having done so in the Intro, Section 2 ('Setting the stage') will be rewritten in a manner that the contents of the subsequent sections 3 to 6 will more logically 'fall in its place' with respect to the above formulated objectives and readers' expectations. Furthermore, where relevant, the introductory text to the different sections will be reformulated or, if necessary, extended to provide additional guidance to maintain the 'bird's eye view' with respect to Section 2.

**Article objectives**.
"I found it difficult to figure out what objectives you have for the article. Early on you talk about your simulation and its purpose of helping students understand CC. However, later, in Section 3, you go on to talk about the workings and adequacy of various climate models, without (I think) clearly showing how the intricate nature of these climate models and what seems to be your suggested model fit into your simulation for students. In my view Section 3 belongs in a journal on climate modelling."

Section 3 in fact focuses on the cryosphere contribution to the multi-scale simulation set-up, the principle objective of my study. It does so by analyzing available paleoclimatic observations from a climate-sensitivity and feedback perspective. Thus, it provides the

necessary link between a possible long-term (cryosphere) contribution and contemporary complex (coupled atmosphere-ocean) climate models. As described above, this should be made more clear to the reader by rewriting the introductory text in relation to the 'Big picture' laid down in Section 2.

"I may be missing something, but I also found it hard to understand how your Sections 4 and 5 relate to your simulation? Section 4 is titled "simulation set-up", which made me expect to see aspects of the practical working of your simulation in the classroom, such as, starting the simulation, interface configuration, student interaction with the simulation, etc. However, I then encountered more discussion about the climate system, and not about your simulation, how it is used, and what results you have obtained in using it. Section 6 seems to do something similar."

Section 4 actually describes a suitable extension of a simple, transfer-function based simulation model (described in more detail in Appendix A), to incorporate the (long-term) cryosphere contribution derived in Section 3. This roughly defines the dynamic simulation core which stands at the basis of the multi-scale simulation model, in the paper referred to as the "simulation set-up".
Section 5 presents a first 'validation' of this combined set-up, by reproducing both 'short' and 'long-term' responses to different possible present-day GHG mitigation scenarios such as provided by the IPCC.
Again, this should be made more clear to the reader, in my opinion feasible by adapting the introductory and concluding text of the sections with this objective in mind.

"Your **Methods** section provides interesting mathematical analysis and model comparison. However, is this the objective of your simulation, and is this what you wish your students to learn? If so, then it does not seem to match (fully?) your objectives as laid out at the start.

All that gives the impression that your accomplished (as opposed to your stated) **objective** is to discuss aspects of climate modelling. Of course, that is of crucial importance, and is discussed in several climate journals, two of which are published by the AGU and the EGU. Again, *Geoscience Communication,* as eclectic and interdisciplinary as it is, does not, in my view stretch that far, unless it is to discuss how such models can communicate aspects of geoscience. May I suggest that you bring out this aspect of your article: How does your tool (better) communicate, and how do we know that it does?"

The purpose of Section 6 is to add a second (analytical) 'validation' to determine how my multi-scale simulation set-up performs in relation to 'new findings' on contemporary estimates of climate sensitivity based on the output of complex GCM's (General Circulation Models) as reported in the recent Nature study of Cox et al. (2018). I agree that this should be made much more clear to the reader in the section intro, again with the original objectives of the study in mind.

"My suggestion is to focus on your simulation as a **learning tool**. For example, you could discuss (and measure and analyse) the various ways in which students benefit from your simulation – how your simulation is revived by, and communicates climate change to, a given audience. You can of course, focus on a (small) selection of (already existing) models, and allow students to manipulate the various variables and feedbacks in the climate system: GHG behaviour, feedback loops (eg, increase in atmospheric $H_2O$, thawing of $CH_4$ substrates), albedo effects, etc. This audience is likely to be a more sophisticated audience."

I agree that this would really be the 'proof of the pudding' of the "STAGE 2.0" endeavour, however at this stage not feasible yet. Once given the opportunity to integrate my model in an actual educational set-up, various options would become available for a proper human-factors evaluation. This could either be of a formative or summative nature, by carefully

designing an experimental set-up in combination with adequate instructional design for a proper measure of 'information/knowledge' transfer to the student for the different models under consideration.

"At one point you say "The main learning objective is to 'get a feel' for both 'short-term' (current century) and possible 'long-term' (beyond) consequences of greenhouse-gas **mitigation** measures." However, I could not find how students in your simulation would manipulate variables representing mitigation measures and thus see their effects."

I agree that at least some additional explanation is required in Sections 4 and 5 on how model inputs were specified to generate the different simulation results presented there. For the implementation in a pedagogical context the very powerful concept of 'direct manipulation' would be recommended, enabling students to manipulate both model inputs (e.g. GHG emissions in GtC/year) and parameters with an immediate feedback of both short- and long-term consequences. The underlying STAGE 2.0 simulation core is exactly designed for this purpose.

"You say that "The conceptual design of the tool is based on the paradigm '**learning as experimenting**', encouraging students to explore climate sensitivity in its various aspects in an active manner.". In my view, this is what your article should focus on, not the detailed analysis of the validity of various climate models. You state that your tool aims to counter common misconceptions [regarding the climate system]. This sounds like it is intended for a lay or general-public audience, not students specializing in climate models. If so, then the mathematics will not serve any purpose. In a simulation, you need to strike a careful balance between detail (a high fidelity simulation) and relatively simple pedagogically-useful simulation. The beauty of pedagogical simulation is its ability to represent reality at the most useful level of representivity for a given audience (sufficiently simple to provide powerful insight and so as not to swamp the main message, and sufficiently complex to provide realistic insight); the more sophisticated (educated) the audience, the more complex the simulation. At a certain point of increasing sophistication, the simulation will cease to have a pedagogical purpose and will manifest a research purpose, poorly adapted to helping lay people understand. Some useful discussion on this has been published, and could be cited."

Regarding the 'common misconceptions' in relation to the level of the intended audience, I admit that I am aiming for an application in a university-level course on climate change. The 'common' refers to that also in this specialistic field misconceptions might be present of a rather persistent nature.

Regarding model complexity I totally agree that this should be determined by the learning objectives for the intended audience, also in relation to my previous comment on specifying model inputs. A typical (complex) GCM (General Circulation Model) as the ones presented in Table C1 used for my analytical STAGE 2.0 model validation (Section 6 of my paper) may take up to months of 'calculation time' (interval between specification of model input to generation of model output), which makes the concept of 'direct manipulation' totally unworkable! Hence, the absolute need for model simplification, which for a qualitative sensitivity study in a pedagogical setting shouldn't cause any real problems with respect to required realism (here the model validation sections in my manuscript come in). The mathematics in my paper is solely intended to derive/describe the simple model in a rigorous manner, as a means of 'information transfer' to potentially interested users (a.o. pedagogical software developers in the field on climate-change education).
I will explicitly address these issues in the introductory sections of my paper, also in relation to your next comment.

"You mention "the paradigm '**learning as experimenting**', encouraging students to explore climate sensitivity in its various aspects in an active manner". You really do need to provide

references to this method. I would probably not call it a paradigm, but rather a method or an approach. A widely used approach goes under the name of the '**experiential learning cycle**', pioneered by Dave Kolb. Simulation designed and conducted within the framework of this cycle tends to provide a powerful method for communication, and is thus relevant to *Geoscience Communication*. However, a word of warning: No game, simulation or role-play or similar activity can hope to produce its full learning potential without the crucial step of **debriefing**. For the purpose of debriefing, it would be useful for facilitators and participants to have the underlying model, probably in non-mathematical terms, for example, as a system dynamics model. One question in the debriefing might cover 'what if' this or that variable in the model were modified?"

I agree, embedding the pedagogical approach of 'learning as experimenting' in existing methods of active learning (Kolb's experiential learning cycle, constructivism/constructionism) deserves more attention in the paper. Again, to my opinion this belongs in the Introduction section, as it provides the pedagogical basis for my STAGE 2.0 model development.

Your suggestion of the *'what-if'* like manipulation of model inputs and parameters directly fits into the objective of designing a proper experimental set-up in case of a human-factors evaluation of the STAGE 2.0 set-up, as mentioned above in my response. This could nicely be complemented with so-called *'how to'* exercises, asking students to determine the required input (e.g. GHG concentration in the atmosphere) in order to achieve a certain output of the climate system (Global surface temperature). This would provide valuable material for an effective debriefing session on how the concept of 'climate sensitivity' is appreciated by the students, also in relation to your final comment:

"I am not sure if you have done this, but a study on the **communicative effectiveness** for the various audiences of your simulation would contribute greatly the literature. Simulation can be a particularly powerful tool for communicating insights into geoscience phenomena, but it and its use (including the debriefing) need to be evaluated in a rigorous manner."

---

## Author Comment (AC2) · 3 Sep 2018

Thank you very much for your time to review my manuscript.

Although you obviously appreciate its main objective by recognizing that "Communicating and conveying the critical thresholds and feedbacks within the climate systems to non-specialists is a laudable objective", you at the end of your review arrive at the recommendation to reject the manuscript for publication in Geoscience Communication.

Your main argument is that to achieve its main objective, summarized as "helping to communicate climate sensitivity to non-specialists", a fundamental re-focusing and re-organisation of the study would be required, which is too large a task to be major revision.

[Figure]

Although I agree with your summary at the end of your review that the present version of the manuscript "does a poor job of explaining who it is for, what its objectives are and what its key messages are", I however see possibilities for fundamental improvement with respect to these issues in a feasible manner. I refer to my more detailed response to the first reviewer's comments, who encountered more or less identical main objections as the ones formulated by you.

––––––––––––––––––––––––––––